# Isometric Representations in Neural Networks Improve Robustness

## Abstract

Artificial and biological agents are unable to learn given completely random and unstructured data. The structure of data is encoded in the distance or similarity relationships between data points. In the context of neural networks, the neuronal activity within a layer forms a representation reflecting the transformation that the layer implements on its inputs. In order to utilize the structure in the data in a truthful manner, such representations should reflect the input distances and thus be continuous and isometric. Supporting this statement, recent findings in neuroscience propose that generalization and robustness are tied to neural representations being continuously differentiable. However, in machine learning, most algorithms lack robustness and are generally thought to rely on aspects of the data that differ from those that humans use, as is commonly seen in adversarial attacks.

During cross-entropy classification, the metric and structural properties of network representations are usually broken both between and within classes. This side effect from training can lead to instabilities under perturbations near locations where such structure is not preserved. One of the standard solutions to obtain robustness is to train specifically by introducing perturbations in the training data. This leads to networks that are particularly robust to specific training perturbations but not necessarily to general perturbations. While adding ad hoc regularization terms to improve robustness has become common practice, to our knowledge, forcing representations to preserve the metric structure of the input data as a stabilising mechanism has not yet been introduced.

In this work, we train neural networks to perform classification while simultaneously maintaining the metric structure within each class, leading to continuous and isometric within-class representations. We show that such network representations turn out to be a beneficial component for making accurate and robust inferences about the world. By stacking layers with this property we provide the community with an network architecture that facilitates hierarchical manipulation of internal neural representations. Finally, we verify that our isometric regularization term improves the robustness to adversarial attacks on MNIST.

## 1 Introduction

Using neuroscience as an inspiration to enforce properties in machine learning has roots dating back to the birth of artificial neural networks (McCulloch & Pitts, 1943; Rosenblatt, 1958). One way to study natural and artificial neural networks is to look at how they transform specific structural properties of input data. The output of such a transformation is typically called a neural, or latent, representation, and it carries information about the computational role of a brain region or network layer (Kriegeskorte, 2008; Kriegeskorte & Diedrichsen, 2019; Bengio et al., 2013). Different properties of representations are helpful in different ways for both organisms and artificial agents. Some examples of this are efficient coding Barlow et al. (1961), mixed selectivity (Rigotti et al., 2013), sparse coding (Olshausen & Field, 2004), response normalization (Carandini & Heeger, 2012), efficiency and smoothness (Stringer et al., 2019) and expressivity (Poole et al., 2016; Raghu et al., 2017) among others.

For example, one subsection of theories related to efficient coding proposes that neural circuits should generate discontinuous and high-dimensional representations to pack the most information

possible into a network (Barlow et al., 1961; Simoncelli & Olshausen, 2001). On the other hand, empirical results point out that neural circuits generate low dimensional smooth representations of the data (Gao & Ganguli, 2015; Gao et al., 2017). This apparent contradiction has already been rigorously discussed in Stringer et al. (2019). According to the work of Stringer et al., neural circuits try to be as efficient as possible while smoothly mapping inputs. Without this smoothness constraint, infinitesimal perturbations of input stimuli could drastically change the output, thereby making such circuits non-robust to some perturbations. Given the empirical support, it seems likely for these properties to hold in early sensory systems and thus to be important for a broad class of machine learning algorithms.

Organisms seem particularly robust to random input perturbations. However, artificial models suffer from a lack of robustness to adversarial attacks Goodfellow et al. (2014). One argument for why this happens can be deduced from Naitzat et al. (2020), in which the authors use a topological approach based on persistent homology (Carlsson, 2009; Edelsbrunner & Harer, 2022), to study the mappings realized by neural networks performing a classification task. They claim that, in classification problems, neural networks implement structure-breaking (non-homeomorphic) mappings, and as argued above, models implementing such mappings are unlikely to be robust. There are many ways to improve the robustness of a network (Madry et al., 2017; Silva & Najafirad, 2020; Xu et al., 2020). Nevertheless, of particular interest to us are strategies that try to solve this problem by restricting the properties of the mapping realized by a network. Examples of this are Jacobian regularization (Hoffman et al., 2019), spectral regularization (Miyato et al., 2018; Nassar et al., 2020), Lipschitz continuity (Virmaux & Scaman, 2018; Liu et al., 2022), topological regularization (Chen et al., 2019) and manifold regularization (Jin & Rinard, 2020) among others.

While regularities in neural representations help with robustness, they do not necessarily guarantee that the input and output representations will have the same metric relationships, thereby reflecting the actual structure of the data. To our knowledge, there are still no methods to preserve the class metric structure while allowing for robust classification. To achieve this behavior, we create a neural network model with, what we call, *Locally Isometric Layers* (LILs) and study the representations generated by training such networks. Furthermore, we extend these networks to generate representations in a hierarchical manner, which makes them helpful in performing classification at different resolutions. Finally, we train LILs on MNIST and show that the isometry condition leads to an improvement in network robustness to both the *Fast Gradient Sign Method* (FGSM) and the *Projected Gradient Descent* (PGD) adversarial attacks.

## 2 BACKGROUND AND METHODS

In this section, we summarise the mathematical background of some of the different mappings that a neural network can implement and introduce LILs. We treat both training and test data as being sampled from a manifold $\mathcal{M}$ and the neural network $\mathcal{N}$ as a set of maps $\mathcal{N} = \{f_i^l | f_i^l : \mathcal{M} \to \mathbb{R}\}$, with $l$ denoting the layer and $i$ - the index of a neuron. Another way to interpret the action of a neural network, which will become useful later on, is to define its mapping layer-wise.

**Definition 1:** *A Neural Network $\mathcal{N}$ with $L$ layers acting on a manifold $\mathcal{M}$ is a set of functions $\{F^l : \mathcal{M} \to \mathbb{R}^{n_l}\}_{l=1}^L$, where $n_l$ is the number of neurons in a particular layer.*

To perform classification, one usually tries to train a network to realize a particular function $\Phi : \mathcal{M} \to \mathbb{R}^{n_L}$, which holds easily separable representations of the data. After this, a simple linear layer can be used for the final classification. This procedure requires the specification of a cost function. Here we will consider the following example:

$$\mathcal{L}(X, T) = T \log \sigma[\Phi(X)] + \frac{1}{N^2} ||G \odot D_{\mathcal{M}} - G \odot D_{\Phi}||_F^2 = \mathcal{L}_{CSE} + \mathcal{L}_{ISO}, \quad (1)$$

where $X = \{x_1, ..., x_N\}$ are the inputs, $T = \{t_1, ..., t_N\}$ are the desired labels, $\sigma$ is the Softmax function, $||\cdot||_F$ is the Frobenius norm, $D_{\mathcal{M}}$ and $D_{\Phi}$ are the distance matrices in the input and output space generated by $d(x_i, x_j)$ and $d(\Phi(x_i), \Phi(x_j))$ respectively and $G$ is an indexing matrix. The distance functions can be any metric, but in this work, we stick to the Euclidean distance. Given a partition of the training set $V = \bigsqcup_k V_k$, $V_k \subset \mathcal{M}$, the indexing matrix $G$ is defined in the following way:

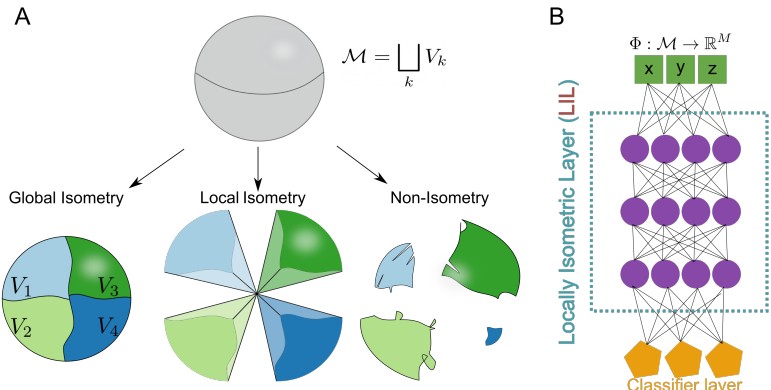

Figure 1: A) Visualization of the three types of mappings. B) The architecture of a *Locally Isometric layer* (LIL). Green nodes correspond to an input layer, purple to hidden layers and orange to a classification layer.

$$G(x_i, x_j) = \begin{cases} 1 & \text{if } x_i,\, x_j \in V_k\,, \\ 0 & \text{otherwise}\,. \end{cases} \qquad (2)$$

This indexing term forces the network to maintain the local distance relationships between the input points in the output layer. The metric preservation is enforced within each class separately, which helps the network find representations that are easy to classify while also preserving the relational information of the data. This second loss term works hand in hand with the first loss term, which is just the usual cross-entropy loss regularly used in classification problems and helps to separate the different classes. This procedure might seem reminiscent of a triplet loss, which is widely used in contrastive learning (Hadsell et al., 2006; Mao et al., 2019). The vital difference is that instead of minimizing the distance between points in the same class, it enforces the distances between them to be the same as in the input data.

In this context, it is also important to mention the work of Sengupta et al. (2018), where the authors imposed a similar metric constraint on ReLU networks to explain the emergence of localized receptive fields. However, they were not attempting to solve classification tasks and, as a result, did not adapt their loss function only to preserve metric relationships within the class. A similar loss function, without the local within-class indexing term $G$, is also one of the main components behind some unsupervised dimensionality reduction algorithms (Tenenbaum et al., 2000), especially those based on multidimensional scaling. The main difference between our method from such algorithms is that they are used to project data to a low-dimensional space in which it can be visualized. Additionally, they are usually not realized by neural networks and are not used for classification.

## 2.1 THREE TYPES OF MAPPINGS

One can describe the mappings of neural networks in many ways (Hornik, 1991; Bianchini & Scarselli, 2014; Guss & Salakhutdinov, 2018; Yang & Salman, 2019), but the main property which is of interest to us is that of metric preservation. Using such a description, we end up with three types of mappings: global isometries, local isometries, and non-isometries. Visual representations of these different mappings are given in figure 1A. A definition of these concepts is in order.

**Definition 2:** *Given a metric space $(X, d)$ and a mapping $F : X \to Y$. $F$ is an isometry (i.e. preserves the metric) if $d(x, y) = d(F(x), F(y))$. If this property only holds locally, meaning $d(x, y) = d(F(x), F(y))$ for $x, y \in V$, where $V \subset X$, we call it a local or piecewise isometry.*

To get these different regimes, we can weigh the loss in the following way,

$$\mathcal{L} = \alpha \mathcal{L}_{CSE} + \beta \mathcal{L}_{ISO}, \qquad (3)$$

with $\beta = 0$, we only maintain the standard cross-entropy term and obtain discontinuous representations. For $\beta > 0$, one gets locally isometric maps. The case with global isometry can be obtained by setting $G_{ij} = 1$, $\forall i, j$ since then, the isometry loss is enforced for all points independent of their class. We will refer to layers whose weights are updated with $\beta > 0$ as *Locally Isometric Layers* (LIL). As mentioned before, by partitioning the data manifold $\mathcal{M}$ into different subsets, the network enforces isometry on different local patches or at different resolutions.

## 2.2 EXPERIMENTS

Our initial experiments were done on relatively low dimensional data so that we could use small and tractable neural networks made up of 4 layers consisting of 20 neurons with hyperbolic tangent activation functions. So for both the entangled rings and torus task, each LIL had the following feedforward architecture - [D,20,20,20,20,C], where D is the dimension of the dataset the LIL receives and C is the number of neurons in the final classifying layer. The networks in the toy datasets were trained for 10000 epochs with a batch size equal to the number of training points using the Adam optimizer (Kingma & Ba, 2014) in PyTorch (Paszke et al., 2019). The choice of a large batch size guarantees that each time the gradient is computed, there will be more points in the same class, and thus the distance relationships between them will contribute more to the loss. If these networks are trained solely by stochastic gradient descent with a batch size of 1, the isometric term will not contribute anything to the loss, which would be equivalent to only using the cross-entropy term.

When training on MNIST, we slightly increased the network size to 100 neurons and did not use any hierarchical structure as that is not present in the labeling of MNIST. The classification was done with a single LIL of 4 layers with 100 neurons each. We used a batch size of 100 for five epochs for the training. To test the robustness obtained by adding the isometric loss term, we trained six networks with the following $\beta$ parameters - [0,0.001,0.01,0.1,1,10]. We performed $L_\infty$ FGSM adversarial attacks with a step size varying logarithmically from 0.01 to 1. For the PGD attacks, we fixed the $L_\infty$ ball size to 0.5 and took ten steps with a varying step size in the same range.

## 3 RESULTS

### 3.1 DISENTANGLING ENTANGLED DATA

Inspired by (Naitzat et al., 2020), we generated a dataset in which the two classes take the shape of two topologically entangled rings in 3-dimensional embedding space, which means that there is no way to separate the two without cutting one of them apart. However, the rings are not entangled in the same way in higher dimensional spaces, as one can move one of the rings in the direction of the fourth dimension, thereby putting it in a different subspace. Given that neural networks project to spaces of much higher dimension, an operation like this should be simple to implement. After training a network with LILs, we find a mapping that disentangles the two manifolds while preserving their topological structure. This solution is very different from the one achieved by only using the cross-entropy loss, where as expected, the topological structure is lost - see figure 2A.

In order to visualize the rings in the network's last layer, we use the UMAP dimensionality reduction algorithm (McInnes et al., 2018). In addition, we show the distance distributions between the points in each ring after passing through the two networks, which heavily supports the idea that the LIL implementation manages to preserve the within-class distance relationships almost perfectly 2B. In contrast, using the cross-entropy loss results in no preservation of structure and instead brings points closer together, as indicated by the exponential shape of the orange histograms, which appears even when a log scale is used.

### 3.2 ISOMETRIC MAPPINGS ARE ROBUST

As seen in the previous section, imposing isometry as a condition on the output layer of a neural network leads to a within-class continuous mapping. We propose that this is a result of the following properties:

**Definition 3:** *A function is called K-Lipschitz continuous if there exists a constant $K \geq 0$, for which* $d(F(x), F(y)) \leq Kd(x, y)$.

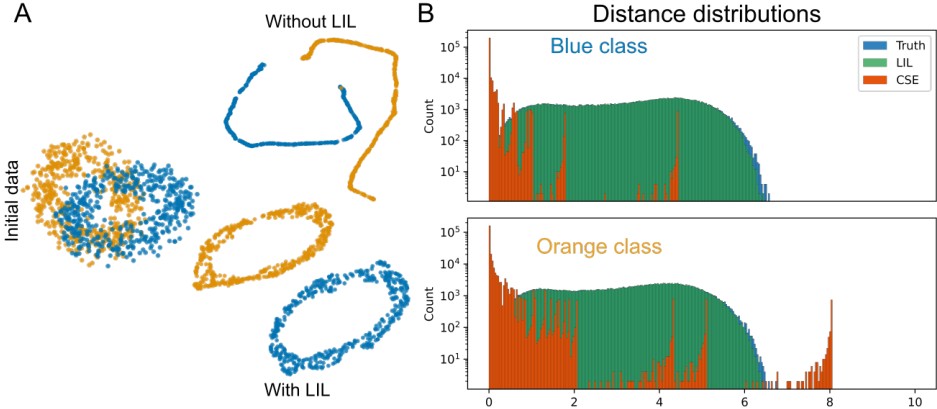

Figure 2: A): The initial dataset and UMAP projections (number of neighbors=40 and minimal distance = 0.25) of the last layer with and without LIL. B): The distance distributions between the embeddings in the last layer for each of the two rings, trained solely with CSE (orange) and with the isometric term included (light green).

With this definition in mind, one can see that:

**Proposition:** Isometric mappings are 1-Lipschitz continuous.

**Proof:** Given the definition of an isometry: $d(x, y) = d(F(x), F(y))$. Simply let $K = 1 + \epsilon$ and obtain the inequality $(1 + \epsilon)d(x, y) \geq d(F(x), F(y))$. The smallest value for which the equality holds is 1, which is the Lipschitz constant.

Such a mapping has additional desirable properties like being almost everywhere differentiable due to Rademacher's theorem (Heinonen et al., 2001) and having derivatives bounded by the Lipschitz constant.

**Proposition:** K-Lipschitz continuous mappings have a bounded derivative.

**Proof:** Rearrange the inequality from the previous proposition to: $\frac{d(F(x), F(y))}{d(x, y)} \leq K$. Then take $x = y + \delta$, this leads to the derivative $||\nabla F||_\infty \leq K$.

All of these properties improve robustness to adversarial attacks since they enforce the gradients of the loss with respect to the data to be bounded. To see this consider the gradient computed in a fast gradient sign method (FGSM).

$$\nabla_x \mathcal{L}(x, t) = \nabla_x \alpha \mathcal{L}_{CSE} + \nabla_x \beta \mathcal{L}_{ISO}. \tag{4}$$

By expanding these terms and applying the chain rule, we obtain the following:

$$\alpha \nabla_x \mathcal{L}_{CSE} = -\alpha \frac{t}{\sigma(\Phi(x_i))} \frac{\partial \sigma(\Phi(x_i))}{\partial \Phi(x_i)} \frac{\partial \Phi(x_i)}{\partial x} \leq -\alpha \frac{t}{\sigma(\Phi(x_i))} \frac{\partial \sigma(\Phi(x_i))}{\partial \Phi(x_i)} K, \tag{5}$$

$$\beta \nabla_x \mathcal{L}_{ISO} = \frac{2\beta G}{N^2} \sum_{i>j} H_{i,j} J_{i,j} [\Phi_i, \Phi_j] \leq \frac{4\beta G}{N^2} \sum_{i>j} H_{i,j} J_{i,j} K. \tag{6}$$

Here we have compressed our notation and redefined the distance matrix difference as $H_{i,j} = d(x_i, x_j) - d(\phi(x_i), \phi(x_j))$, the difference between the mapping at two different points as $J_{i,j} = \frac{\phi(x_i) - \phi(x_j)}{d(\phi(x_i), \phi(x_j))}$ and the difference of the gradient evaluated at two different points as $[\Phi(x_i), \Phi(x_j)] = \nabla_x \Phi(x_j) - \nabla_x \Phi(x_i)$. To see a more detailed derivation of equation 6, see Appendix A.1.

Thus, by exploiting the isometric property, one can enforce neural networks to implement mappings with a bounded derivative, which is expected to improve robustness because it keeps gradients with respect to the data small.

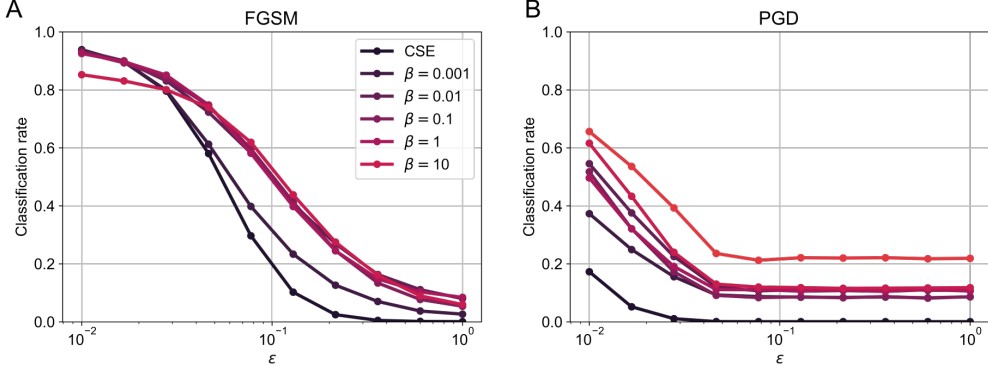

Figure 3: A) Robustness of a LIL to an FGSM adversarial attack. B) Robustness to a PGD adversarial attack with a $L_\infty$ bounding neighborhood of size 0.5.

## 3.3 ROBUSTNESS TO ADVERSARIAL ATTACKS ON MNIST

To show this empirically, we trained the LIL model on MNIST, as described in 2.2. We see a stark improvement in robustness to both FGSM and PGD attacks as the constant $\beta$ controlling the importance of the isometry condition is increased. The results are shown for FGSM and PGD in figure 3, panels A and B, respectively. However, this improvement does not come without a cost; see table 1, showing that increasing the importance of isometry leads to a degradation in terms of performance. At the same time, smaller values keep the performance around the cross-entropy baseline. Thus models which perform mappings very close to isometry are much more robust, but they pay the price for that by having sub-optimal performance in the absence of perturbations.

Table 1: MNIST test performance as a function of the $\beta$ parameter which controls to what extent the isometry property contributes to the loss.

| $\beta=0$ | $\beta=0.001$ | $\beta=0.01$ | $\beta=0.1$ | $\beta=1$ | $\beta=10$ |
|-----------|---------------|--------------|-------------|-----------|------------|
| **0.9701** | 0.9688 | 0.9686 | 0.9665 | 0.9588 | 0.8816 |

## 3.4 LOCALLY ISOMETRIC LAYERS AND HIERARCHICAL REPRESENTATIONS

Another way to use LILs is by stacking them in order to glue or cut subsets of the data manifold, as shown in figure 4A. The first LIL implements some function $\Phi : \mathcal{M} \to \mathbb{R}^{n_L}$, which splits the original manifold into $C$ submanifolds (corresponding to the number of classes) while maintaining the distances between the points. After the first split is performed, the following LIL operates on the representation already split by the previous layer, giving a new partition $\mathcal{M} = \bigsqcup_k V_k$. Depending on the choice of the partition, each following LIL can be used to either glue or cut out pieces of the original manifold. For the gradients not to interfere, we impose the condition that for each LIL, gradients are backpropagated only to their preceding LIL, but this assumption can be relaxed. In this work, we also use a shared classification layer, for which we only need to train the linear projection from each LIL. However, the same result can be achieved by appending separate classification layers to each LIL.

In this final example case, we use this stacked LIL model to look at a simple torus split into cylindrical regions 4B. We sampled 1600 points from the standard three-dimensional parametrization of the torus and added a small amount of noise $\nu \, \mathcal{N}(0, 0.001)$ to it. We set the $\beta = 100$, which is relatively high. We used only one batch, consisting of all the sampled points from a torus. This way, the isometric condition is applied fully in all epochs. This is a rather simple example case, and the network has to perform the task by separating the different regions of the torus into locally isometric submanifolds - figure 4. Because, in this case, we make use of the hierarchical version of our model, the regions of the torus are partitioned with different degrees of coarseness across

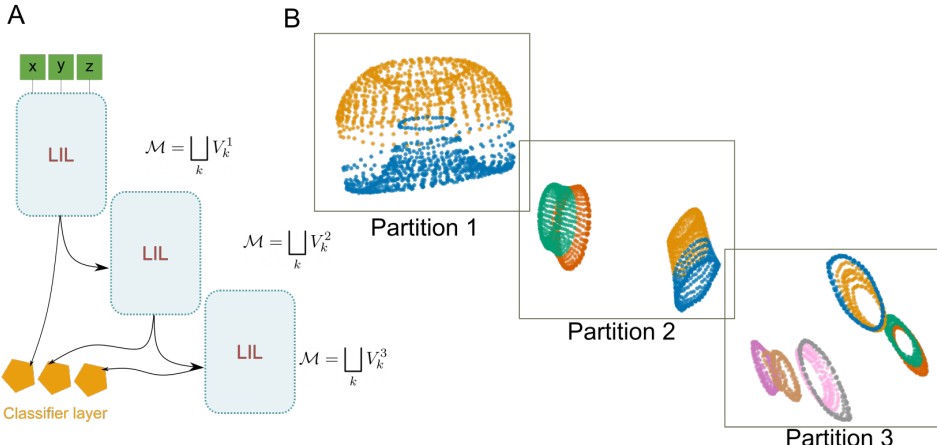

Figure 4: A) Stacked LILs, implementing different locally isometric maps depending on the partitioning (labeling) of the input data. B) UMAP visualizations (number of neighbors = 40, minimal distance = 0.25) of the representations in a classification task in hierarchically stacked LILs.

layers. The stacked LIL network manages to separate the classes well while maintaining a high classification rate ($> 0.99$).

## 4 DISCUSSION

### 4.1 ADVANTAGES OF ISOMETRIC MAPPINGS

In this work, we developed a method for classification while maintaining the local metric relationships between points in the input data. This type of mapping is beneficial for the truthful representation of structural relationships in data and for improved robustness to targeted perturbations like those used in adversarial attacks. We used a Euclidean metric to represent the structural relationships between data points. However, the approach can also be extended to other metrics more suitable to the data being analyzed. In our toy dataset experiments, we show that the LIL implementation completes a classification task without destroying the local structure within each class.

At the same time, standard neural networks trained with solely cross-entropy perform non-homeomorphic mappings. This result contrasts the reasoning in Naitzat et al. (2020), which argues that non-homeomorphic mappings are necessary to disentangle topologically entangled data, simplifying the topology of the data and allowing for separation by hyperplanes. Using LILs, comparable classification performance is made possible while the topology and even the metric structure of the initial data manifold are preserved within each class throughout the network. From a mathematical perspective, this is expected as one can disentangle data in higher-dimensional spaces obtained with a projection of wide neural network layers.

### 4.2 ROBUSTNESS TO ADVERSARIAL ATTACKS

Our results on MNIST indicate a clear improvement in robustness to adversarial attacks. One possible explanation for this improvement is that the isometry condition imposes a small Lipschitz constant on the function mapping data or stimuli to neural representations. This makes the derivatives of such mappings bounded. As a result, perturbation strategies that rely on computing gradients with respect to data, like those employed in FGSM and PGD, will be weaker. One might also use similar reasoning to explain the increased robustness observed when adding either smoothness or more direct Lipschitz constraints, as seen in Chen et al. (2019); Hoffman et al. (2019); Jin & Rinard (2020); Nassar et al. (2020); Liu et al. (2022), among others.

On the other hand, it is unclear what the Lipschitz constant of a traditional cross-entropy trained neural network is, despite attempts to provide an estimate (Virmaux & Scaman, 2018). If such

networks turned out to have an even smaller Lipschitz constant almost everywhere except at a few breaking points where the topological structure is not maintained, they would still be expected to be robust. In that case, there would have to be a different explanation for the improvement in robustness that we observe.

We also point out that using the LIL strategy with large $\beta$ values decreases classification performance. This implies that, at least in this example case, there is a trade-off between performance and robustness. This trade-off has also been found in other work (Tsipras et al., 2018) and might be a necessity or a feature of how we achieve robustness. In any case, it is an interesting problem to explore in future work.

### 4.3 HIERARCHICAL SPLITTING OF REPRESENTATIONS

Finally, we have also allowed our model to take advantage of hierarchical structure in labeled data. Hierarchical classification has been attempted previously (Srivastava & Salakhutdinov, 2013; Deng et al., 2014; Wehrmann et al., 2018; Scieur & Kim, 2021). However, to our knowledge, this has not been done while also maintaining the local within-class structure of the data. At least intuitively, by mapping the data to the first class partition in a locally isometric manner, one guarantees that the following more fine-grained classification network will work with a truthful representation of the data. While we have not explored this intuition further in this work, we believe that this is something that would be interesting to investigate in hierarchically structured datasets like Imagenet (Krizhevsky et al., 2017) or the many of the other publicly available and more specialized alternatives (Wei et al., 2021) in the future. Another possible application of these concepts is in the area of dimensionality reduction and data visualization. For example, one can use hand-labeling or unsupervised hierarchical clustering algorithms to obtain plausible data labels. Given such labeling, LILs can be used to project the data to a low dimensional space in which the different classes are separated while the metric structure is maintained.

## 5 CONCLUSION

To summarize, we have extended the existing regularization methods by introducing a local isometry condition that keeps the structure within classes consistent across network layers. Nevertheless, such a mapping still allows for different classes to be separated, which is a necessary feature of classification. Additionally, we have explored some peculiar features of this type of regularization. For one, due to the isometry property, it is capable of continuously untying entangled data. We have also shown that in virtue of such properties, gradients with respect to the input data are bounded, which is expected to improve robustness to adversarial attacks. This is empirically demonstrated in simulations on MNIST. Finally, we propose that our LIL model can be stacked to improve hierarchical classification.

ACKNOWLEDGMENTS

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

# A    APPENDIX

## A.1    DERIVATION FOR LIPSCHITZ BOUND ON THE ISOMETRIC LOSS GRADIENT.

We start be writing down the isometric loss term:

$$\mathcal{L}_{ISO} = \frac{\beta}{N^2}||G \odot D_{\mathcal{M}} - G \odot D_{\Phi}||_F^2. \tag{7}$$

When taking the gradient we can rewrite it in terms of the distance generating functions $d(x_i, x_j)$ and $d(\phi(x_i), \phi(x_j))$. Furthermore for simplicity of notation we will abbreviate $\Phi(x_i)^l$ to $\phi_i^l$, where $l$ is one of the basis components (eg. a pixel of an image). Lastly for the purpose of readability, we will omit writing out the element-wise multiplication operator for the indexing matrix G. This gives the following:

$$\beta\nabla_x\mathcal{L}_{ISO} = \frac{\beta G}{N^2}\nabla_x \sum_{i,j}(d(x_i, x_j) - d(\phi_i, \phi_j))^2. \tag{8}$$

By expanding the squared brackets and applying the chain and product rules, the expression becomes:

$$\frac{2\beta G}{N^2}\sum_{i,j}[d(x_i, x_j)\nabla_x d(x_i, x_j) - d(x_i, x_j)\nabla_x d(\phi_i, \phi_j) -$$
$$d(\phi_i, \phi_j)\nabla_x d(x_i, x_j) + d(\phi_i, \phi_j)\nabla_x d(\phi_i, \phi_j)], \quad (9)$$

rearranging these terms we get:

$$\frac{2\beta G}{N^2}\sum_{i,j}\{d(x_i, x_j)[\nabla_x d(x_i, x_j) - \nabla_x d(\phi_i, \phi_j)] + d(\phi_i, \phi_j)[\nabla_x d(\phi_i, \phi_j) - \nabla_x d(x_i, x_j)]\} =$$
$$\frac{2\beta G}{N^2}\sum_{i,j}\{[d(x_i, x_j) - d(\phi_i, \phi_j)][\nabla_x d(x_i, x_j) - \nabla_x d(\phi_i, \phi_j)]\}. \tag{10}$$

In order to continue it helps to compute the derivatives of the distance functions with respect to the basis in which the input data is described, this gives:

$$\nabla_x d(x_i, x_j) = \frac{x_i^l - x_j^l}{d(x_i, x_j)}e^l,$$
$$\nabla_x d(\phi_i, \phi_j) = \frac{\phi_i^l - \phi_j^l}{d(\phi_i, \phi_j)}\frac{\partial\phi(x_i)}{\partial x^l}e^l, \tag{11}$$

with $e^l$ being the standard basis components.

Substituting these expressions and taking advantage of the symmetry in the distance functions we rewrite the expression as:

$$\frac{2\beta G}{N^2}\sum_{i>j}\{[d(x_i, x_j) - d(\phi_i, \phi_j)]\sum_l[\frac{x_i^l - x_j^l}{d(x_i, x_j)}e^l - \frac{\phi_i^l - \phi_j^l}{d(\phi_i, \phi_j)}\frac{\partial\phi(x_i)}{\partial x^l}e^l]$$
$$+ [d(x_i, x_j) - d(\phi_i, \phi_j)]\sum_l[\frac{x_j^l - x_i^l}{d(x_i, x_j)}e^l - \frac{\phi_j^l - \phi_i^l}{d(\phi_i, \phi_j)}\frac{\partial\phi(x_j)}{\partial x^l}e^l]\}, \tag{12}$$
$$= \frac{2\beta G}{N^2}\sum_{i>j}[d(x_i, x_j) - d(\phi_i, \phi_j)]\sum_l(\frac{\phi_i^l - \phi_j^l}{d(\phi_i, \phi_j)})(\frac{\partial\phi(x_j)}{\partial x^l} - \frac{\partial\phi(x_i)}{\partial x^l})e^l.$$

At this point it helps to compress our notation further. We redefine the difference between the two distance matrices as $H_{i,j} = d(x_i, x_j) - d(\phi_i, \phi_j)$, the difference between the mapping at two different points as $J_{i,j} = \frac{\phi_i - \phi_j}{d(\phi_i, \phi_j)}$ and the difference of the derivative at two different points as $[\phi_i, \phi_j] = \nabla_x \phi_j - \nabla_x \phi_i$ (inspired by the notation used in Lie brackets). Given these new identifications we obtain the final form of the gradient as:

$$\beta \nabla_x \mathcal{L}_{ISO} = \frac{2\beta G}{N^2} \sum_{i>j} H_{i,j} J_{i,j} [\phi_i, \phi_j]. \tag{13}$$

Given that for an isometry the norm of the gradient $||\nabla_x \phi|| \le 1$, the difference between the gradients evaluated at any two points is bounded by $2K$. With this in mind we arrive at the final inequality, showing that the isometric loss gradient is bounded:

$$\beta \nabla_x \mathcal{L}_{ISO} = \frac{2\beta G}{N^2} \sum_{i>j} H_{i,j} J_{i,j} [\phi_i, \phi_j] \le \frac{4\beta G}{N^2} \sum_{i>j} H_{i,j} J_{i,j} K. \tag{14}$$

$\square$

