# OpenReview forum: "Isometric Representations in Neural Networks Improve Robustness"
_ICLR.cc/2023/Conference — Submitted to ICLR 2023_

### Official Review · Reviewer_RnHw · 2022-10-18

**Confidence:** 5
**Correctness:** 3
**Technical Novelty And Significance:** 2
**Empirical Novelty And Significance:** 2
**Recommendation:** 3

**Clarity, Quality, Novelty And Reproducibility:**

The paper is clearly written. However, the overall novelty is suspect as there have been several works which leverage local lipschitzness (see minor concern above.). The authors have shared code and results, and therefore have satisfied any reproducibility concerns. THe overall quality of the paper is however not up to the mark, given the lack of experiments on harder datasets, and testss with more powerful attacks.

**Strength And Weaknesses:**

**Strengths**
1. The paper is well motivated with principled arguments about why the locally isometric network should work.
2. The writing is concise, clear, and well organized.

**Weaknesses**

1. The evaluation is very sparse compared to the standards accepted by the community. Most adversarial robustness papers test their approach on a variety of networks, and datasets. In addition, the attacks used here are not enough to ensure robustness (see Tramer et al (On adaptive attacks to adversarial example defenses), Carlini et al. (On evaluating adversarial robustness) for a more detailed discussion).
2. In addition, the paper does not present any comparisons with corresponding methods (adversarial training, TRADES, MART) which all present counters to PGD type attacks.
3. Another issue would be scalability. While the synthetic data and MNIST are mostly well separated, real world image data is both high dimensional and non-separable. My conjecture is that the second term of the loss will lead to significant accuracy losses for CIFAR-10 and Imagenet. I suggest the authors update the paper with these experiments if possible.

Minor concern: While the idea of enforcing local Lipschitzness is inspired, it is not new. Several works [1,2,3] have approached the problem of adversarial robustness by constraining local robustness. The actual implementation differs in each case, however, the overall approach is similar.
Refs:
1. Mao et al., Metric Learning for Adversarial Robustness, Neurips 2019
2. Finlay et al, Lipschitz regularized Deep Neural Networks generalize and are adversarially robust
3. Yang et al, Adversarial Robustness Through Local Lipschitzness

**Summary Of The Paper:**

The paper presents an isometry aware training approach that improves robustness. The main intuition is that if the latent representations (pre final layer in this case), are (locally) Lipschitz, the network will be robust to adversarial perturbations. The algorithm enforces this Lipschitz constraint using a regularizer which penalizes any deviation in pairwise distance matrices of the outputs and inputs. The paper presents evidence for this in form of experiments on several synthetic datasets (constructed with specific topographic properties) and MNIST.

**Summary Of The Review:**

Overall, the paper needs some additional work -- especially with better experiments -- to be acceptable. While local Lipschitzness does imply robustness, it is really hard to execute for larger datasets. I am willing to change my opinion if the authors show some evidence of their approach succeeding on CIFAR-10.

---

### Official Review · Reviewer_78Vo · 2022-10-24

**Confidence:** 5
**Correctness:** 2
**Technical Novelty And Significance:** 2
**Empirical Novelty And Significance:** 2
**Recommendation:** 3

**Clarity, Quality, Novelty And Reproducibility:**

Clarity: Good

Quality: Borderline

Novelty: Good

**Strength And Weaknesses:**

Strength:

1. The Locally Isometric Layer (LIL) is proposed to avoid the structure-breaking property in neural network classification, which sounds interesting.

2. The proposed method is clearly presented.


Weaknesses:

1. The reason why LIL makes the NN more robust is not well-explored. In Section 4.2, there is a discussion on the relationship between Lipschitz constant and the LIL, but no empirical study is provided to compare the performance of LIL to other methods that also reduces Lipschitz constant, such as [1,2,3].

2. The experiment is only carried out with MNIST, which is a quite simple dataset. To show the effectiveness of LIL, the experiment on CIFAR10 is at least needed.

3. The experiment result on MNIST is quite poor if compared with existing benchmarks such as [4]. The robust accuracy under $\epsilon=0.3$ constraint is over 80%, while Fig. 3 only shows an accuracy of ~20% when $\epsilon=0.3$, which is a huge gap. Thus, I seriously doubt the effectiveness of LIL.

[1] Chao Chen, Xiuyan Ni, Qinxun Bai, and Yusu Wang. A topological regularizer for classifiers via persistent homology. In The 22nd International Conference on Artificial Intelligence and Statistics, pp. 2573–2582. PMLR, 2019.

[2] Judy Hoffman, Daniel A Roberts, and Sho Yaida. Robust learning with jacobian regularization. arXiv preprint arXiv:1908.02729, 2019

[3] Hsueh-Ti Derek Liu, Francis Williams, Alec Jacobson, Sanja Fidler, and Or Litany. Learning smooth neural functions via lipschitz regularization. arXiv preprint arXiv:2202.08345, 2022

[4] https://github.com/MadryLab/mnist_challenge

**Summary Of The Paper:**

The paper proposes to learn isometric representations in neural networks to preserve the structure in the output end. Then locally isometric layer is proposed to learn the required output representation, which is shown to be robust on MNIST compared with the cross-entropy training baseline.

**Summary Of The Review:**

The paper has a major flaw in experiment as mentioned in the Weaknesses, so I tend to reject the paper.

---

### Official Review · Reviewer_1G3T · 2022-10-24

**Confidence:** 3
**Correctness:** 2
**Technical Novelty And Significance:** 1
**Empirical Novelty And Significance:** 2
**Recommendation:** 3

**Clarity, Quality, Novelty And Reproducibility:**

The paper is overall clear.
The novelty is somewhat incremental.
The results are a bit premature, and the basic idea may need be to investigate in more detail.

**Strength And Weaknesses:**

Strength: The idea is relatively straightforward and intuitive. Some preliminary experimental results are provided.

Weakness:
However, the results in the current form are too premature. There is also a fundamental concern with the premise of the proposed approach.
Making the representation isometric will certainly increase the robustness to perturbation in the input as it directly constrains the Lipschitz continuity. However, this does not touch on the key issue of robustness. Making a neural network robust to image classification tasks requires the network to be sensitive to the on-manifold perturbations (those that change the perceptual properties, e.g., the identity of an object), while being insensitive to the nuisance variability (e.g., increasing the contrast of the stimulus). For the approach proposed by the authors, increasing the contrast of the stimulus or a global scaling of the image would pose a major challenge to the network for solving the image classification task. The preliminary results in Table 1 are consistent with this interpretation and there is a corresponding drop in the performance of the network (as acknowledged by the authors).

Other comments:
The second paragraph in the Introduction is a misinterpretation of the existing neuroscience literature. The results of Stringer et al are inconsistent with the classic efficient coding idea. The representation they proposed is closer to the “fractional” representation. Traditional neural population coding models with broadly tuning neurons will create representations that are more consistent with the isometric idea. The paper below seems to contain an additional neuroscience example that might be useful for motivating the study: Gao et al. "On Path Integration of Grid Cells: Group Representation and Isotropic Scaling." Advances in Neural Information Processing Systems 34 (2021): 28623-28635. But they’re dealing with a low-dimensional problem, i.e., physical space.


**Summary Of The Paper:**

This paper proposes to regularize neural networks using isometric properties. The paper presents some intriguing ideas along with some preliminary results that would be worth exploring further.

**Summary Of The Review:**

This is a submission with an intriguing idea. The results are premature.


**** Edits after the rebuttal:

I appreciate the authors' responses to my critiques.  In its current form, I consider the study to be premature. Performing additional research along the directions the authors raised in their response will likely make the study stronger.

---

### Official Review · Reviewer_aCCN · 2022-10-24

**Confidence:** 4
**Correctness:** 2
**Technical Novelty And Significance:** 2
**Empirical Novelty And Significance:** 2
**Recommendation:** 3

**Clarity, Quality, Novelty And Reproducibility:**

The paper is clear explain and somewhat novel. I offer a suggestion of a possibly relevant paper to contrast with.


**Strength And Weaknesses:**

The paper is well written, and the problem is very interesting to the community. The clarity made the paper enjoyable to read.


The paper, however, falls short in various aspects.

1) the experimental evaluations are too sparse to evaluate the models. It is tested on one dataset per task. As mentioned earlier, this paper is reminiscent of that of Nøkland et al. 2019, but not cited in the paper.
2) the models consider have various parameters, and none of their influences are thoroughly studied.
3) as mentioned by the authors, competing models exist, yet the current model is not compared to those.

The idea is clear and interesting, but the claims are not well-supported.



**Summary Of The Paper:**

The authors propose to combine the existing neural network's loss with an isometry-preserving loss, which they argue would allow for preserving the geometry, which the authors complement with some theoretical results.

The two main experiments aim to show that the resulting model is more robust to adversarial attacks and leads to better accuracy when the geometry is preserved across layers.

The paper is somewhat reminiscent of [1], combining similarity-preserving objectives and standard objective function layer-wise. The empirical results nonetheless are very promising but too sparse to be of real value.

[1] Nøkland, Arild, and Lars Hiller Eidnes. "Training neural networks with local error signals." International conference on machine learning. PMLR, 2019.


**Summary Of The Review:**

The paper is interesting and the current experiments are ok. However, there is a lot that is missing to be able to actually confirm the claims made by the authors. We would like more thorough evaluations of the models. It is too premature to be accepted as a publication given that it is not possible to clearly evaluate the approach.

---

### Author Response · Authors · 2022-11-19
**Response to reviewers**

We wish to thank the reviewers for their insightful comments and recommendations. We agree that our paper would greatly benefit from a stronger verification of the method on different architectures and datasets. Since we are too limited on time to do a proper rigorous verification of the requested modifications to our method, we will implement the suggested additions in a future version of the paper. Even though we have, at the moment, not performed a complete verification we still believe the method is valuable and its main features are shown to be interesting and worthy of publication. For completeness, we summarize and respond to the main critiques raised by the reviewers.

__1. The method will fail to scale up to more complex datasets in which the data is both high-dimensional and inseparable.__
- We agree that such datasets offer additional challenges and will accordingly adapt the paper to make note of such difficulties. As an example, pixelwise Euclidean distances might not be an informative description of the relations between images, but the proposed isometric approach is independent of the chosen initial metric. Therefore, one can use any data-appropriate metric along with our approach, in which case achieving good performance is partially a matter of working with the right metric. In the future we plan to show results on CIFAR10 and make note of the dependence on the chosen metric.

__2. There is no comparison of our approach to some of the existing methods for adversarial robustness.__
 - While we agree that such a comparison is both interesting and relevant, the reason we avoided this was mostly because we are not aiming to outperform the state of the art in terms of robustness. Rather we wanted to explore how the property of isometry determines the nature of neural representations. This is especially pertinent to biological neural networks, which are unlikely to follow very hand-crafted implementations, but could follow a more general principle like having a small Lipschitz constant. In the future we will extend the text to make our intentions more clear and add the performance of the other mentioned methods.

__3. There is no exploration of the performance dependence on the parameters of the model__.
 - While it is difficult to obtain a clear understanding of this dependence, we agree that such a comparison is necessary and will be performed in a future version of the paper.

__4. When it comes to image classification one should focus on remaining robust specifically to on-manifold perturbations. Furthermore, nuisance variables, like contrast or features irrelevant to the classified object, should be ignored. (raised by 1G3T)__
 - This is a good point with which we strongly agree, however our approach should be interpreted more generally. While one of our experiments is done in the context of image classification (MNIST), the approach can be applied to any setting in which one can identify a reasonable metric. For example if one picks a metric which is invariant to changing the contrast of an image, then such perturbations will likely not be misclassified. In the future iteration of this work we will adapt the text of the manuscript to underline the generality of our approach.

__5. The second paragraph of our manuscript is a misrepresentation of the neuroscience literature. (raised by 1G3T)__
 - Since we agree with the reviewer that the work of Stringer et al is at odds with efficient coding, we are unsure why what we are saying is a missrepresentation. The main point we wanted to make in this section was that representations should be somewhat smooth and our piecewise isometric approach will capture such smoothness while breaking it when necessary. Notice that our claim is not that the Lipschitz constant is globally bounded, but rather that it is only bounded within each class. If anything one would expect that around the decision boundaries the representations generated through our approach are highly non-smooth and thus close to the fractional representations that the reviewer mentioned.

__6. We have missed to cite similar work (Nøkland et al). (raised by ACCN)__
 - We thank the reviewer for pointing out this work to us, as we were unaware of it. While the implementation and motivation in that work are slightly different, we agree that this is a highly relevant example to include and compare with. As a result we will cite and discuss this work in the new version of our manuscript.

---

> ### Comment · Reviewer_aCCN · 2022-11-24
> **Response to authors**
>
> Thank you for taking the time to address some of the comments in your response.
>
> I appreciate that the authors "agreed" to explore the avenues the reviewers have pointed out, but it has not been implemented in a new version of the manuscript. As of now, I cannot change my score based on the current state of the paper.

---

> ### Comment · Reviewer_RnHw · 2022-11-30
> **Response to the Authors**
>
> Thank you for your clarifying statements. In terms of robustness, I understand that the authors are not aiming to beat state-of the-art. However, the point of evaluating with the right attacks is to accurately (or as accurately as possible) model the robustness of the proposed representations. Since that is a property claimed in the paper, I am still of the opinion that the evaluation is incomplete.
>
> I am also unable to change my score given the current manuscript.

---

> ### Comment · Reviewer_78Vo · 2022-12-02
> **Comment**
>
> My concern on the experiment comparison is apparently not addressed and I am also unable to change my score given the current manuscript.

---

### Decision · Program_Chairs · 2023-01-20

**Decision:**

Reject

**Justification For Why Not Higher Score:**

Even though the paper presents an interesting approach for enforcing isometric mapping in classification models, unfortunately, all the reviewers have found the paper premature in its current form.

**Justification For Why Not Lower Score:**

N/A

**Metareview: Summary, Strengths And Weaknesses:**

This paper explores isometric representation learning and its impact on adversarial robustness.

Pros:
- Clarity in presentation
- Interesting direction for adversarial robustness is explored

Cons:
- Poor experimental setup (small toy datasets)
- Lack of comparison against major competing methods

**Summary Of Ac-Reviewer Meeting:**

N/A